# The Diversity Metrics of Sub-models based on SVD of Jacobians for Ensembles Adversarial Robustness

**Ruoxi Qin[1], Linyuan Wang[1], Xuehui Du [2] Bin Yan[1]*†Xingyuan Chen[2]†**

[1]Henan Key Laboratory of Imaging and Intelligent Processing, PLA Strategy Support Force Information Engineering University
[2]PLA Strategy Support Force Information Engineering University
18530023930@163.com, wanglinyuanwly@163.com, dxh37139@sina.com, ybspace@hotmail.com, chxy302@vip.sina.com
Science Road, Suite 62
Zhengzhou, Henan 450001

## Abstract

Transferability of adversarial samples under different CNN models is not only one of the metrics indicators for evaluating the performance of adversarial examples, but also an important research direction in the defense of adversarial examples. Diversified models prevent black-box attacks relying on a specific alternative model. Meanwhile, recent research has revealed that adversarial transferability across submodels may be used to abstractly express the diversity needs of sub-models under ensemble robustness. Because there was no mathematical description for this diversity in earlier studies, the difference in model architecture or model output was employed as an empirical standard in the assessment, with the model loss as the optimization aim. This paper proposes corresponding assessment criteria and provides a more accurate mathematical explanation of the transferability of adversarial samples between models based on the singular value decomposition (SVD) of data-dependent Jacobians. A new constraints norm is proposed in model training based on these criteria to isolate adversarial transferability without any prior knowledge of adversarial samples. Under the novel condition of high-dimensional inputs in training process, the model attribute extraction from dimensionality reduction of Jacobians makes evaluation metric and training norm more effective. Experiments have proved that the proposed metric is highly correlated with the actual robustness of transferability between sub-models and the model trained based on this constraint norm improve the adversarial robustness of ensemble.

## Introduction

In the research of adversarial examples, transferable adversarial examples have become an important research direction because of their more flexible and extensive application scenarios in practice(Akhtar and Mian 2018). As a way to improve the robustness, ensemble has become an important research direction to defense against adversarial samples at this stage. Essentially, the robustness of ensemble model is due to the well-calibrated uncertainty estimation for adversarial samples that outside the train-

*Corresponding authors.
†These authors contributed equally.

ing data distribution(Lakshminarayanan, Pritzel, and Blundell 2016). Related test results combined with research (Kuncheva and Whitaker 2003) proposed the concept of diversity of sub-models under ensemble conditions, and experimentally demonstrated that the robustness of the ensemble has a certain correlation with the diversity of sub-models.

Ensemble is widely used on both sides of attack and defense in related competitions, and the description of diversity metric is summarized as the diversity of model structure(Kurakin et al. 2018). More studies have proved that models trained on the same data set without additional constraints are more inclined to extract the same non-robust features (Ilyas et al. 2019; Li et al. 2015) making such an empirical defense method not always effective in practice.More research hopes to further define the diversity between models through an abstract characterisation, so as to obtain sub-models based on diversity constraint and improve the robustness of the ensemble(Bagnall, Bunescu, and Stewart 2017; Pang et al. 2019; Kariyappa and Qureshi 2019; Yang et al. 2020). The common problem of these methods is that the define of diversity is only based on abstract concepts without mathematical description, so its evaluation is more restricted from the perspective of optimization loss.

Based on the conclusion of the correlation between transferability and diversity of sub-models(Yang et al. 2020), this paper proposes an metric for accurately evaluating model diversity based on the SVD of the Jacobian matrix. And through the singular value and vector from mathematical metric, the above-mentioned abstract expression is further explained theoretically. Geometrically, Figure 1 simplify demonstrate the difference between the evaluation method proposed in this paper and methods based on abstract characterisation by the level set of the optimize problem gradient, and gives a more accurate definition of transferability in theory. Further, a regular term constraint through the proposed diversified evaluation metric is used in model training process to generate diversified sub-models, thereby improving the robustness of ensemble. In summary, the main contributions of this article are as follows:

- This paper proposed a quantitatively metric based on SVD of the Jacobian matrix for adversarial transferability.

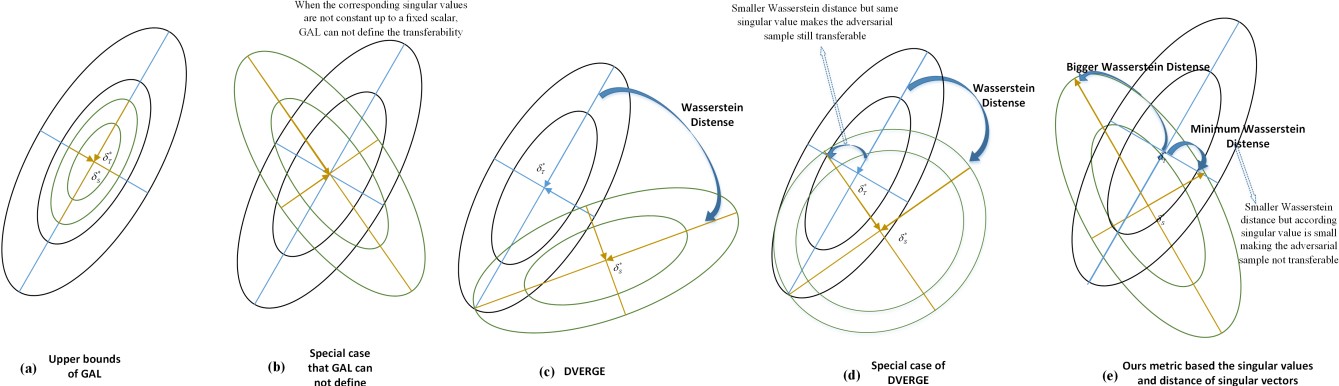

Figure 1: The illustration of the different metric of transferability based on level set of the optimize problem. (a) The upper bound defined in GAL; (b) Transferability disturbance conditions that cannot be accurately defined by GAL; (c) DVERGE maximizes the distance between the optimal disturbances to limit transferability; (d) Transferability disturbance conditions that cannot be accurately defined by DVERGE; (e) The definition of transferability in this paper based on singular value and Wasserstein distance of singular vector.

- The mathematical characterisation based on optimization theory of the transferability further help us to understand model attribute of the black-box.
- This paper further uses the metric of diversity metric as a regular norm item in network training so as to improve the ensemble robustness.

## Related work

**Abstract description and hypothesis of sub-model diversity** Subsequent studies proceed from different assumptions and put forward different evaluation metric for the diversity of sub-models under ensemble robustness. Based on the difference non-maximal logits outputs of model, ADP(Bagnall, Bunescu, and Stewart 2017; Pang et al. 2019) evaluated such diversity between sub-models. Based on the overlap of adversarial subspaces (Tramèr et al. 2017), GAL(Kariyappa and Qureshi 2019) evaluates such diversity through the gradient direction difference; Based on the non-robust features, DVERGE(Yang et al. 2020) further evaluated such diversity through the non-robust distill feature transferability. Different from the above-mentioned assumption, this paper starts from the perspective of the transferability of adversarial samples on the basis of the assumption of (Yang et al. 2020), and gives a mathematical expression through optimization theory.

**Theoretical analysis of model attributes based on Jacobian matrix** The attributes of the model for more theoretical analysis is the essential to explain the black-box performance.The Frobenius norm of the Jacobian matrix is first used in the regularization training of the model's robustness (Hoffman, Roberts, and Yaida 2019; Jakubovitz and Giryes 2018; Novak et al. 2018). When the adversarial samples are initially discovered, the spectral norm of a certain layer weight is considered to be a metric for evaluating the sensitivity (Szegedy et al. 2013). The global spectral norm of the model Jacobian matrix is further used to con-

strain the robustness of the model (Sokolić et al. 2017; Farnia, Zhang, and Tse 2018). (Khrulkov and Oseledets 2018; Roth, Kilcher, and Hofmann 2019) essentially reveals that the iterative generation process of adversarial samples is mathematically approximates to the SVD of Jacobian matrix through power method (Boyd 1974). Through further mathematical analysis, the Frobenius norm of Jacobian matrix is connected with the transferability of Universal Adversarial Perturbations (UAP) (Co, Rego, and Lupu 2021).

This paper expands the theoretical analysis based on Jacobian matrix and innovatively evaluate the transferability between models accurately through SVD. The Jacobian matrix after dimensionality reduction is decomposed, and the degree of alignment between the singular vectors is precisely defined by the Wasserstein distance.

## Method

Define $f(x)$ as the logit output of convolutional neural network $f$ under image $x$, while $J_f(x) = \frac{\partial f_i}{\partial x} |_x$ is the Jacobian matrix under image $x$. In the case where the perturbation $\delta$ is small enough and higher-order terms are ignored, the degree variation output of model which measured by the regular term $L_q$ can be linearly represented as the first-order Taylor expansion through Jacobian matrix $J_f(x)$:

$$\|f(x+\delta) - f(x)\| \approx \|J_f(x)\delta\|_q \le \|J_f(x)\|_F \|\delta\| \quad (1)$$

From the perspective of optimization theory, the goal of adversarial sample optimization is to maximize $\|J_f(x)\delta\|_q$. When $q = 2$ the goal of adversarial sample optimization can be simplified to a constrained optimization problem of quadratic functions:

$$
\begin{aligned}
maxmize \quad & \delta^T Q \delta \\
subject \quad to \quad & \delta^T P \delta = K
\end{aligned}
\quad (2)
$$

Where $Q = J^T J$. Because of the homogeneity of the norm, k is set as 1 to solve this constrained optimization problem.

Through Lagrange function $l(x, \lambda) = x^T Q x + \lambda(1 - x^T P x)$ under constrained optimization problem, the Lagrange condition can be obtained as:

$$P^{-1} Q \delta = \lambda \delta \quad (3)$$

Therefore, the eigenvector of $P^{-1}Q$ is the optimal $\delta$ that corresponding to the solution of objective equation (3). When the perturbation constraint is also under $L_2$norm, $P$ is the identity matrix, and the maximum eigenvalue of $Q$ is the maximum value of equation (3). It can be seen that the singular vector of the Jacobian matrix $J$ essentially defines the possible local optimal solutions of $\delta$, and the maximum singular value defines the maximum output variation of the model under $L_2$norm. Without this meaning of singular values, the upper bound of transferability was defined through inequality in (1)(Kariyappa and Qureshi 2019). But as shown in Figure 1(a) and (b), when singular vectors are not fullly align and singular values are not constant up to a fixed scalar this metric can not define the transability more accurately. Through the theory of optimization, the meaning of eigenvalues and eigenvectors can be combined to further analyze the transferability.

In order to evaluate the transferability of adversarial samples more accurately, this paper characterize the transferability based on the distance between singular vectors. How to choose a reasonable distance function is an essential issue in our method. (Gulrajani et al. 2017) proposed theory that the constraint of the variation between the logites output of different images is essentially a constraint on the Wasserstein distance of the image. Converting to the scenario of adversarial distinguish, the diversity metric in DVERGE can be expressed as the discrimination of GAN to distinguish the adversarial samples. So the diversity constraint achieved by the DVERGE increase the distance between optimal perturbations. As shown in Figure 1(d), considering extreme cases when singular values of a target Jacobian matrix is not much different and there is one target singular vector has small Wasserstein distance with source optimal perturbations, it can also achieve strong transferability under this constraint. A more accurate assessment of transferability defined in this paper is characterized as: Given the singular vector($s\_vec$) corresponding to the largest singular value of the source Jacobian matrix($max(s\_val_{J_s})$), the singular value($s\_val$) corresponding to the target Jacobian singular vector under the condition of minimizing the Wasserstein distance($mindis\_s\_val_{J_s \to J_t}$) reveals the approximate output variation. Let $d$ as Wasserstein distance, the Equation (4) mathematically expresses this metric as:

$$\frac{mindis\_s\_val_{J_s \to J_t}}{max\left(s\_val_{J_s}\right) \times min\left(d\left(\underset{s\_vec_{J_s}}{argmax}\left(s\_val\right), s\_vec_{J_t}\right)\right)} \quad (4)$$

$$s.t. \quad mindis\_s\_vac_{J_s \to J_t} = \underset{s\_vec_{J_t}}{argmin}\, d\left(\underset{J_s}{argmax}(s\_val), s\_vec_{J_t}\right)$$

Drawing idea from PCA's dimensionality reduction, we make redundant assumptions about the role of batch-size and image-channel dimensions in the gradient, and reduce the dimensionality of the Jacobian matrix through HOSVD decomposition (Kolda and Bader 2009; Chen and Saad 2009). This paper follows the overall parameter optimization of ensemble in the training process. Algorithm 1 shows the overall optimization algorithm.

---

**Algorithm 1:** Ensemble network optimization based on transferability metric

---

**Input**: Batch images X, N sub-models
**Parameter**: Parameters of sub-model
**Output**: models for ensemble

1: initialization or pretraining model reload.
2: **while** i=1..N **do**
3:     Randomly initialize sub-model $f_i$
4: **end while**
5: **while** epoch=1..M **do**
6:     **while** i=1..N **do**
7:       $ens\_out+ = softmax\left(model_i(X)\right)$
8:       **while** j=1..N **do**
9:         $trans\_metrics_i+ = trans\_metrics_{i,j \neq i} \triangleleft eq(4)$
10:     **end while**
11:     $ens\_transs = mean_N(trans\_metrics_i)$
12:     $ens\_loss = BCE\left(mean_N(ens\_out), Y\_onehot\right)$
13:     $g_\omega = \nabla_\omega(ens\_loss + ens\_trans)$
14:     $\omega = \omega + \alpha \cdot RMSProp(\omega, g_\omega) \triangleleft gradient \quad regular$
15:     **end while**
16: **end while**
17: **return** Diversity sub-models

---

## Experiment and results

### Experiment of different evaluation metrics

The experiment combines the evaluation metric described by abstract concepts to discuss the correlation between it and the evaluation metric proposed by this paper and verifies ours effectiveness. DVERGE (Yang et al. 2020) characterizes the degree of output variation of distillation adversarial examples between different sub-models as equation (5):

$$\frac{1}{2} E_{(x,y)(x_s,y_s)}\left[l_{f_i}\left(x'_{f_l^j}(x, x_s, y)\right) + l_{f_j}\left(x'_{f_l^i}(x, x_s, y)\right)\right] \quad (5)$$

Based on the diversity evaluation metric of formula (5), the experimental result in Table 1 shows the diversity evaluation results of different method. The result shows the distillation adversarial loss between sub-models based on formula (5). The brackets after each method give the transferability evaluation metric based on formula (4). The feature distillation of adversarial examples is based on the method of article (Ilyas et al. 2019). The perturbation strength is set as standard 0.03 ($\approx$ 8/255) while the iteration step is 50.

| Ours(4.917) | | | DVERGE(19.757) | | | Baseline(157) | | |
|---|---|---|---|---|---|---|---|---|
| 19.07 | 26.57 | 13.12 | 0.71 | 15.971 | 16.33 | 0.355 | 4.42 | 4.035 |
| 27.689 | 23.63 | 13.17 | 16.438 | 0.82 | 16.28 | 5.08 | 0.39 | 4.46 |
| 27.43 | 26.68 | 9.175 | 16.45 | 15.949 | 0.787 | 5.059 | 4.81 | 0.314 |
| ADP(69.873) | | | GAL(31) | | | Advt(48.565) | | |
| 1.31 | 4.73 | 4.41 | 2.559 | 6.369 | 6.46 | 3.966 | 4.55 | 4.6 |
| 4.59 | 1.197 | 4.517 | 6.07 | 1.448 | 19.48 | 4.68 | 3.86 | 4.598 |
| 4.646 | 5.071 | 1.226 | 6.048 | 17.59 | 1.24 | 4.687 | 4.56 | 3.88 |

Table 1: The diversity evaluation results of different method. The brackets after each method give the metric based on equation (4)

Comparing the results of different methods on different evaluation metric, the results obtained based on equation (4)

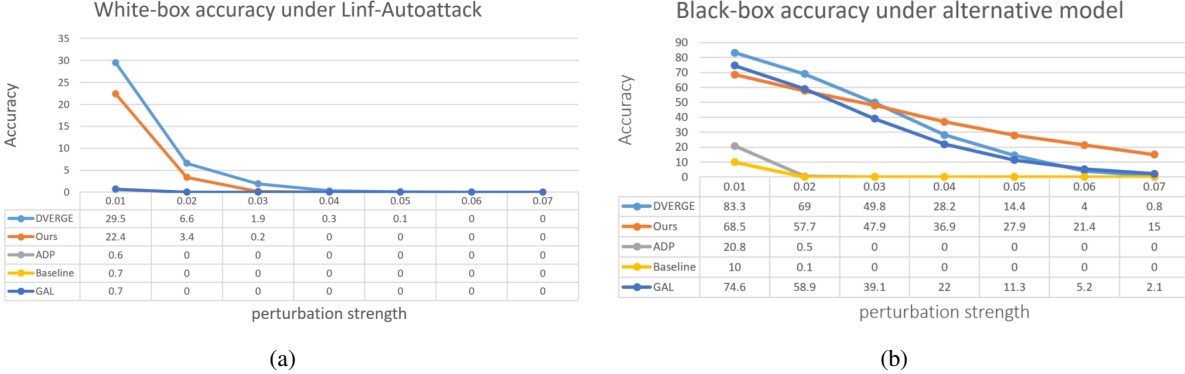

(a)                                                                   (b)

Figure 2: Robustness result with different perturbation: (a)white-box attack; (b)black-box attack. Different line shows the different method to diversify sub-model. All ensemble is achieved under three sub-models.

proposed in this paper are consistent with the results of equation (5) in the evaluation of model diversity. It is demonstrated that the metric proposed in this paper based on equation (4) have coherence in evaluating the output variation caused by perturbation with equation (5). The evaluation metric of equation (4) is based on the attributes of the model itself, and does not depend on any prior information based on adversarial samples. This is also the essential difference between the method in this paper and DVERGE. Also based on the attribute extraction attribute of the Jacobian matrix, the GAL method(Kariyappa and Qureshi 2019) optimizes the upper bound constraint defined by formula (1), which also improves the diversity of the network. However, compared with the metric proposed in this paper, the poor results fully demonstrate that the method in this paper is a more effective characterization of the model's output variation under the transfer attack.

**Experiment of ensemble robustness** The experiment in this section evaluates the robustness of the ensemble model with different method. In order to evaluate the robustness more comprehensively, different adversarial perturbation experiments are set up. Figure 2 shows the according result under white-box attack and black-box attack. The white box attack algorithm uses the current AutoAttack algorithm (Croce and Hein 2020) which has the best attack performance. The black box attack mainly relies on the transfer attack algorithm of the alternative model, which is consistent with the setting of DVERGE. Based on the baseline models, three types of adversarial example including (1) PGD (2) M-DI2-FGSM (Xie et al. 2019) (3) SGM (Wu et al. 2020) are generated, and the final accuracy rate is calculated comprehensively under different types of adversarial samples.

Comparison under the white-box attack, the method in this paper achieves the best robust performance without the any adversarial sample prior conditions. Compared with the optimal result of DVERGE, because the equation (4) only gives constraints from the output variation range, the final recognition accuracy is not characterize enough, so the optimal result is not achieved under the robustness of recogni-

tion, which is the further direction to improve. Comparison under the black-box attack, our method achieves the best defense performance under high perturbation. The robustness under low perturbation conditions is not optimal. By comparing the accuracy of each type of adversarial sample, the adversarial sample based on CW loss has relatively good attack performance. This also shows that the perturbation constraint characterized by the output variation in this paper is still more sensitive to the change of the loss function in practice. The theoretical characterization based on the loss function is an important point to further improve the robustness.

## Conclusion

In this paper, the transferability of adversarial samples between sub-models is taken as the starting point for the study of ensemble robustness. Through the optimization theory analysis under Lagrange conditions, the SVD of the Jacobian matrix is a characterization of the model's optimal perturbation and output variation. Based on this theory, the level set of optimization further mathematical demonstrate the shortcomings of the previous abstract characterization of trasferability. So, this paper effectively redefines the transferability metric between models: Given the singular vector corresponding to the largest singular value of the source Jacobian matrix, the singular value corresponding to the target Jacobian singular vector under the condition of minimizing the Wasserstein distance reveals the approximate output variation. By performing SVD on the dimensionality-reduced Jacobian matrix, the sub-models obtained by this metric as a regular term in network training has a great effect on reducing the degree of output variation. Without relying on any prior information of adversarial samples, experiments show that the method, using as a model attribute extraction, finally improves the robustness of the ensemble. The theoretical characterization of the loss function and classification performance instead of the output variation will be an important direction to further improve the robustness of classification.

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
