# OpenReview forum: "The Diversity Metrics of Sub-models based on SVD of Jacobians for Ensembles Adversarial Robustness"
_AAAI.org/2022/Workshop/AdvML — AAAI-22 AdvML Workshop ShortPaper_

### Official Review · Reviewer_yGxj · 2021-12-01
**Review of Paper 14**

**Rating:** 7
**Confidence:** 4

**Review:**

This paper proposes an SVD-based metric to measure the adversarial transferability between models, based on which the authors further develop an ensemble-based defense, which outperforms previous baselines.

There are some suggestions for extending this paper to a full version:
1. Include AutoAttack in evaluation, which is the most commonly used benchmark attack right now.
2. Polish the format of equations, e.g., Equation (4) where the subscript is not correctly compiled.
3. Figure 1 is informative but is a little bit dazzled to parse.

---

### Decision · Program_Chairs · 2021-12-01

**Decision:**

Accept (Short Paper)

**Comment:**

The reviewer agrees to accept this paper. Please address the reviewer's comment in the camera-ready version.